# Men’s Perceptions of a Gender-Tailored eHealth Program Targeting Physical and Mental Health: Qualitative Findings from the SHED-IT Recharge Trial

**DOI:** 10.3390/ijerph182412878

**Published:** 2021-12-07

**Authors:** Ryan J. Drew, Philip J. Morgan, Frances Kay-Lambkin, Clare E. Collins, Robin Callister, Brian J. Kelly, Vibeke Hansen, Myles D. Young

**Affiliations:** 1School of Education, College of Human and Social Futures, University of Newcastle, Callaghan, NSW 2308, Australia; ryan.drew@newcastle.edu.au (R.J.D.); philip.morgan@newcastle.edu.au (P.J.M.); 2School of Medicine and Public Health, College of Health, Medicine and Wellbeing, University of Newcastle, Callaghan, NSW 2308, Australia; frances.kaylambkin@newcastle.edu.au (F.K.-L.); brian.kelly@newcastle.edu.au (B.J.K.); 3School of Health Sciences, College of Health, Medicine and Wellbeing, University of Newcastle, Callaghan, NSW 2308, Australia; clare.collins@newcastle.edu.au; 4School of Biomedical Sciences and Pharmacy, College of Health, Medicine and Wellbeing, University of Newcastle, Callaghan, NSW 2308, Australia; robin.callister@newcastle.edu.au; 5School of Health and Human Sciences, Southern Cross University, Coffs Harbour, NSW 2450, Australia; vibeke.hansen@scu.edu.au; 6School of Psychological Sciences, College of Engineering, Science and Environment, University of Newcastle, Callaghan, NSW 2308, Australia

**Keywords:** behavior change, depression, males, mental fitness, obesity, online, physical activity, process evaluation

## Abstract

Despite increasing rates of co-morbid depression and obesity, few interventions target both conditions simultaneously, particularly in men. The *SHED-IT: Recharge* trial, conducted in 125 men with depressive symptoms and overweight or obesity, tested the efficacy of a gender-tailored eHealth program with integrated mental health support. The aims of this study were to examine the perceptions of men who received the *SHED-IT: Recharge* intervention in relation to recruitment, satisfaction with the program, and suggestions to improve the program. Individual semi-structured interviews were conducted in a random sub-sample, stratified by baseline depression and weight status (*n* = 19, mean (SD) age 49.6 years (11.6), PHQ−9 score 9.0 (3.7), BMI 32.5 kg/m^2^ (4.6)). Transcripts were analyzed using an inductive process by an independent qualitative researcher. Four themes emerged, namely, (i) specific circumstances determined men’s motivation to enroll, (ii) unique opportunity to implement sustained physical and mental health changes compared to previous experiences, (iii) salience of the program elements, and (iv) further opportunities that build accountability could help maintain focus. Gender-tailored, self-directed lifestyle interventions incorporating mental health support are acceptable and satisfying for men experiencing depressive symptoms. These findings provide important insights for future self-guided lifestyle interventions for men with poor physical and mental health.

## 1. Introduction

Depression and obesity are highly prevalent, chronic health conditions that affect a substantial number of men worldwide [1,2]. Evidence indicates depression and obesity commonly co-exist [3], with a meta-analysis detecting a complex reciprocal pathway between the two conditions [4]. Of additional concern, depression and obesity are associated with heightened risks for other health complications (e.g., hypertension, type 2 diabetes) [5] and substantially reduced life expectancy [6,7]. Although effective treatments to manage these conditions exist, men remain less likely than women to access psychotherapy or participate in lifestyle modification programs [8,9], including weight loss interventions [10]. Of concern, some men even perceive these options to be unappealing and contrary to traditional views of masculinity [11]. For these reasons, it is important to increase the range of effective and appealing treatment approaches for men seeking to improve their physical and mental health.

Lifestyle programs integrating specific mental health support can reduce weight and depressive symptoms concurrently [12], but most data are from studies with predominantly female participants [12,13]. However, when men are targeted in gender-tailored behavior change programs that account for male interests and values [14], engagement has been high with important physical (e.g., weight loss) and mental health (e.g., quality-of-life) benefits observed [15,16]. Compared to conventional lifestyle programs, gender-tailored approaches integrate socio-culturally relevant characteristics in the design and delivery of health behavior interventions that men prefer (e.g., relatable facilitator, sensitive use of humor) [17]. This reinforces the concept that men are willing to adopt healthy behavior changes if they find the program suitable and tailored to their preferences [16,18]. Notably, programs that focus largely on lifestyle behaviors, including increasing physical activity, may be particularly appealing for men who align with dominant forms of masculinity [19] and/or those who are dismissive of their depressive symptoms [20]. Importantly, lifestyle programs targeting physical and mental health have been effectively delivered via eHealth approaches [21,22]. According to Pagliari and colleagues [23], eHealth interventions are defined as those that “deliver health services through the internet and related technologies” (e.g., mobile apps, websites, online modules). The online nature of these programs (e.g., confidentiality, flexibility) may be a preferred option for men who are unwilling to engage in face-to-face support [24], while also aligning to traditional masculine norms such as self-reliance and independence [20].

To address the growing number of men living with both depression and overweight or obesity, we recently conducted the *SHED-IT (Self-Help, Exercise and Diet using Information Technology): Recharge* randomized controlled trial (RCT). This study evaluated the efficacy of a 3-month gender-tailored, eHealth program comprising behavioral lifestyle advice with targeted mental health support to improve the physical and mental health of men with overweight or obesity and low mood. The program was completely self-guided and included printed materials (e.g., a logbook to document key program tasks) alongside online components (e.g., interactive online modules, study website). All materials were gender-tailored to appeal to men’s preferences (e.g., scientific legitimacy, relatable male-only information, online delivery mode) [17]. Detailed results on the primary outcomes of depressive symptoms and weight [25], secondary measures (i.e., health behaviors and cognitions) [26], and mediators of depressive symptoms [27] have been reported elsewhere. Compared to the wait-list control group, the *SHED-IT: Recharge* group maintained significant intervention improvements at 6 months follow-up for depressive symptoms (adjusted mean difference: −2.4 Patient Health Questionnaire (PHQ-9) units, 95% CI = −4.0, −0.7, *p* < 0.01, *d* = 0.5) and weight (adjusted mean difference: −3.6 kg, 95% CI = −5.1, −2.1, *p* < 0.001, *d* = 0.8). Small-to-medium intervention effects were also detected for several health behaviors (moderate-to-vigorous physical activity (MVPA), energy intake (kJ)) and cognitive outcomes (cognitive flexibility, behavioral activation) (range, *d* = 0.3–0.5) [26].

Despite these positive findings, a comprehensive process evaluation of the trial, has not yet been conducted. To guide further program developments and provide insights for new programs, it is important to seek feedback from participating men about the design and delivery of the intervention, as well as their experiences in the trial [28,29]. With little evidence of lifestyle programs targeting men with low mood and overweight or obesity [15], process evaluation data can provide key insights for future program development and research, particularly when programs involve several components [28]. Further, qualitative research methods have particular utility for identifying those aspects of a program that participants found engaging and effective [10]. Therefore, the aims of the current study were to examine the perceptions of men who participated in the *SHED-IT: Recharge* trial in relation to recruitment, satisfaction with program content and delivery components, and suggestions for how the *SHED-IT: Recharge* program could be improved.

## 2. Materials and Methods

### 2.1. The SHED-IT: Recharge RCT

A comprehensive description of the *SHED-IT: Recharge* RCT and findings of the study outcomes have been reported elsewhere [25,26,27]. Briefly, 125 men (intervention, *n* = 62; control, *n* = 63) participated in a parallel-group RCT comparing an intervention to a self-assessment, wait-list control group. Assessments were held in September 2019 (baseline), December 2019 (3 months, primary time-point), and March 2020 (6 months, follow-up) at the University of Newcastle, Australia. All measures were collected by trained research assistants who were blinded to group allocation. The trial was approved by the University of Newcastle Human Research Ethics Committee and registered with the Australian New Zealand Clinical Trials Registry (ACTRN12619001209189). All men provided written informed consent prior to enrolment. The study design, conduct and reporting adhered to the CONSORT guidelines [30].

The 3-month self-directed *SHED-IT: Recharge* program was based on the rigorously evaluated *SHED-IT Weight Loss Program for Men*, which provides evidence-based strategies for men to lose weight through practical lifestyle changes [31,32,33,34]. The program contained a range of behavior change techniques linked to key constructs from Bandura’s Social Cognitive Theory (SCT) (e.g., self-efficacy, goal setting) to facilitate sustainable lifestyle change [35]. Consistent with previous *SHED-IT* interventions, the current program was gender-tailored to appeal to men [17]. For instance, the resources contained both surface-structure components (e.g., male-specific statistics, images of men engaging in healthy activities) and deep-structure components (e.g., trustworthy information, humor, realistic communication) that men value [14,17,36]. Further tailoring was linked to the program’s self-directed, online mode of delivery as men are often expected to be self-reliant and independent [37], which may suppress their involvement in one-to-one or group-based programs. In particular, the online nature of the *SHED-IT: Recharge* program was designed to appeal to men by providing efficient access, privacy, and autonomy [19,24]. This was in contrast to previous face-to-face programs requiring ongoing appointments, which have reported lower engagement in men [10]. 

Given the current study targeted men with low mood, the previous *SHED-IT Weight Loss Program for Men* program was modified to specifically focus on improving both physical and mental health simultaneously. This included adapting previously recommended health advice (e.g., physical activity options, diet quality), adding new behavioral recommendations (e.g., sleep, resistance training) linked to depression [38,39], and providing targeted mental health support as a novel component to compliment the behavioral advice [15]. An extensive summary of the intervention components (e.g., recommended tasks) and alignment to theoretical concepts (e.g., SCT for behavioral change, CBT for targeting mental health) is published elsewhere [25]. In brief, intervention components included a combination of eHealth (e.g., website, resistance training videos, mental fitness modules) and printed resources (e.g., program handbook and logbook). To begin, men received a *Handbook*, which contained information on men’s health, weight loss, goal setting, and tips to improve key behaviors (e.g., physical activity, resistance training, sleep, diet quality). Men were also given a *logbook*, where they completed key setup tasks (e.g., review introductory video), set and reviewed goals, mapped their weight loss, worked through online activities (e.g., mental fitness modules, resistance training videos), and documented their progress through the program (e.g., food and exercise diaries). The study website hosted introductory materials (e.g., *SHED-IT Guide to Weight Loss* video) and a series of 10−15 min *resistance training workout* videos. These weekly pre-recorded sessions guided men through a sequence of bodyweight resistance exercises (e.g., squats, push-ups) that were designed to be achievable (exercise variations to make it relevant for the different fitness levels), challenging (difficulty increased weekly), and practical (completed anywhere). As a novel component, targeted mental health intervention through a sequence of *online mental fitness modules* using cognitive behavioral techniques (e.g., cognitive restructuring, behavioral activation) was also integrated due to its effectiveness for improving depressive symptoms [40]. These self-guided modules were designed to be interactive (e.g., brief quizzes) and required follow-up tasks in the logbook. To monitor their daily dietary intake and physical activity, we encouraged men to use the freely available *MyFitnessPal* website or app, which contains a database of food items (including nutrient values) that can be logged to provide individual feedback in relation to weight loss goals (e.g., daily kJ target). Men also received weekly text messages to emphasize key strategies from the program or to direct them to the latest mental fitness or resistance training workout.

### 2.2. Study Design

This study used descriptive qualitative methods to gain a comprehensive understanding of the experiences of men who participated in the *SHED-IT: Recharge* program [41]. Individual semi-structured interviews were conducted between 15 June and the 3 July 2020, which was approximately 3 months after the conclusion of the RCT (and 6 months since men completed the 3 month post-program assessment). Although men were interviewed during the very early stages of the COVID−19 pandemic, participants were specifically asked to reflect on their experiences during the program and the subsequent follow up period (which occurred prior to the pandemic). Given men can be reluctant to discuss mental health concerns [9] and some questions may have been of a sensitive nature (e.g., seeking help when in distress), appropriate interview methods (e.g., one-on-one, suitable time, online) were used to encourage men to provide a more open response [42]. Using the Zoom Video Communications Inc. (Zoom) conferencing program, the online aspect of the interviews provided flexibility from both a participant (e.g., accessibility, convenience) and researcher perspective (e.g., reduced costs, time-efficient), whilst maintaining notable advantages of face-to-face interviewing (e.g., greater personal interface, meaningful participant/researcher rapport) when discussing personal matters [43]. This project was approved by the Human Research Ethics Committee at the University of Newcastle. Men were provided with an additional information sheet explaining the interview aspect of the study and a consent form. All men provided written informed consent prior to their participation in the interview. Participants were informed that their responses would be treated confidentially and anonymized for reporting. 

### 2.3. Participants

Men were eligible to participate in the *SHED-IT: Recharge* RCT if they were aged 18−70 years, had a body mass index (BMI) in the overweight or obesity range (BMI 25−42 kg/m^2^), and reported at least mild depressive symptoms over the previous two weeks (PHQ-9 score ≥5) [44]. For inclusion, men also completed an exercise screening questionnaire or provided medical approval from their general practitioner if any health concerns were detected in the pre-exercise screen. Full eligibility criteria are listed elsewhere [25]. Men were recruited for the *SHED-IT: Recharge* RCT from the Hunter Region of NSW, Australia, between August and September 2019, predominately through traditional news mediums and targeted Facebook advertising. 

Overall, 46 intervention participants provided consent to a post-program interview. This represented 94% of men in the intervention group who attended the 6-month follow-up assessment (*n* = 49) and 74% of the entire intervention group (*n* = 62). We determined that 20 men would provide a sufficient representation of the different subgroups of men in the study. It was achievable in relation to logistical and budget constraints and aligned with sample sizes used in other papers of this nature [32,36]. To ensure a comprehensive representation, we used purposive sampling to stratify the consenting men by baseline weight status (overweight/obesity) and depression status (PHQ-9 symptoms mild/moderate-to-severe). The purpose of this process was to ensure a diverse range of participants were represented, rather than to identify differences in perceptions between groups in the thematic analysis. Twenty men were randomly selected to participate in the interviews using a random number generator in Microsoft Excel, with 19 (95%) providing consent. One participant could not be contacted, despite several attempts via phone and email. The representation from each category was: (i) obesity/moderate-to-severe depression (*n* = 5); (ii) obesity/mild depression (*n* = 5); (iii) overweight/moderate-to-severe depression (*n* = 4); and (iv) overweight/mild depression *(n* = 5). The following measures were collected at baseline in the RCT through online surveys and in-person assessments. Depressive symptoms were measured using the PHQ-9 depression scale, which has established validity as a diagnostic measure of depressive disorders in both clinical and community samples [44]. Weight was measured objectively by research assistants on a digital scale to 0.01 kg (CH−150kp, A&D Mercury Pty Ltd., Perth, Australia). Sociodemographic characteristics (e.g., age, marital status, post-school qualifications) were assessed using a series of questions. Socioeconomic status was determined by linking the participant’s residential postcode to the Index of Relative Socioeconomic Advantage and Disadvantage from the census-based Socio-Economic Indexes for Areas (SEIFA) database [45].

### 2.4. Semi-Structured Interviews

Men from the intervention group participated in individual semi-structured interviews to discuss their experiences of the *SHED-IT: Recharge* program. The interview questions were developed by research team members with expertise in weight loss, mental health, physical activity, nutrition, psychology, and men’s health, within both face-to-face and self-guided interventions. To create the interview questions for this study, we first sourced the interview guides used in recent process evaluations of several health behavior change interventions targeting men [32,36]. After combining these guides, all chief investigators participated in a brainstorming session where the previous questions were either removed, modified, or added to suit the unique areas of interest for the current intervention and sample group (e.g., interaction between physical and mental health). The interview guide covered key areas such as: recruitment, satisfaction with program content and delivery components, suggestions for improvement, and the impact of the *SHED-IT: Recharge* program (see Table 1). Given the comprehensive nature of the interview guide, a variety of important themes emerged. To ensure that all themes can be discussed in sufficient detail, this paper focused on those relating to why men enrolled in the program, their satisfaction with the program, and suggestions to improve the program. Themes relating to the how men perceived the program impacted their physical and mental health will be discussed in a subsequent paper. Probing techniques to facilitate elaboration on responses were also used. This included specific encouragement to help men feel comfortable about expressing any negative views. All transcripts were de-identified prior to the analysis to ensure participant confidentiality. All interviews were conducted by one researcher (R.J.D.) using the same interview guide. Although the intervention was self-directed, the researcher was known to participants through interactions at the study assessment sessions. 

### 2.5. Data Analysis

Using IBM SPSS Statistics version 25 (IMB Corp., Armonk, NY, USA), baseline characteristics of men who were interviewed versus men not interviewed were compared using independent *t*-tests for continuous variables and chi-squared (χ^2^) tests for categorical variables. All variables were checked for plausibility and missing values. Data are presented as mean (SD) for continuous variables and counts (percentages) for categorical variables. 

Audio and video recorded interviews were conducted using the online Zoom Video Communications Inc. (Zoom) conferencing program; however, only the audio recordings were sent for transcription. Interviews were independently transcribed verbatim, with men de-identified prior to analysis. Analyses were conducted using Nvivo 11 (QSR International Pty Ltd., Doncaster, Australia) by author V.H., who is an experienced qualitative researcher and was not involved in any other aspect of the trial. A thematic analysis technique as described by Braun and Clarke [46] was chosen as it was deemed important for theme generation to be a largely inductive process that would allow for an exploratory identification of all aspects of the participant experience. Initially the dataset was examined for overarching themes and a preliminary coding scheme was generated. During the subsequent coding cycle, all data relevant to answering the overall research questions were coded, with further expansion and refinement of the coding scheme taking place. A thematic structure was proposed and refined further after discussion with the research team. The dataset was organized into themes and sub-themes such that these together were deemed to tell the story of the entire dataset. Thematic summaries were developed, which included the integration of representative quotes across the entire dataset. 

## 3. Results

### 3.1. Participant Characteristics

Table 2 demonstrates that the baseline characteristics of men interviewed and men who were not interviewed were comparable. The mean (SD) age of the 19 men interviewed was 49.6 (11.6) years, and mean weight was 103.4 (17.0) kg. The mean PHQ-9 score was 9.0 (3.7), with 53% of men reporting scores in the mild range (5−9) and 47% categorized as in the moderate-to-severe range (10+). At baseline, 42% of interviewed men were accessing some form of mental health support (anti-depressant medication, psychotherapy, or both). Most were born in Australia (90%), had post-high school qualifications (90%), were married or in a domestic relationship (74%), and were employed full-time (69%). There were no statistically significant differences in baseline characteristics (i.e., age, employment status, SES, weight, depression severity) between the men interviewed and those who were not (all *p* > 0.05).

### 3.2. Analysis of Interviews

The average interview duration was 43 min (ranging from 21 to 73 min). Thematic analysis of the interviews revealed a range of themes representing men’s perceptions and experiences of the *SHED-IT: Recharge* program. Four themes emerged: (i) specific circumstances determined men’s motivation to enroll, (ii) unique opportunity to implement sustained physical and mental health changes compared to previous experiences, (iii) salience of the program elements, and (iv) further opportunities that build accountability could help maintain focus. The related sub-themes are listed in Table 3, which are elaborated upon in the sections below. Representative quotes are provided throughout with pre-program age, BMI (kg/m^2^), and PHQ-9 score provided in brackets. 

### 3.3. Specific Circumstances Determined Men’s Motivation to Enrol

For many participants, the motivation to join the *SHED-IT: Recharge* trial was due to long-term *dissatisfaction with their weight*, including frustration with fluctuating weight loss/regain experienced because of previous weight loss attempts. Many talked of a growing difficulty in maintaining an ideal weight as they aged due to perceived lack of physical activity, fitness, or health problems.


*My weight gained quickly, and I found it very difficult to get the motivation to change my eating and drinking habits and get back into that groove of exercise which I found so easy before. So, I was hoping that would be a kick-start for me.*
(ID 11, 59 years, BMI: 32.2 kg/m^2^, PHQ-9 score: 13) 

In terms of the actual event or situation that prompted the shift from contemplation to action, about half the participants had been *prompted by their partner* who had become aware of the program through advertising. 

The *university-based affiliation* of the program was a prominent motivator to enroll for many participants, providing them with reassurance that the program was evidence-based and non-profit. This was important for many with past (and unsuccessful) experiences with profit-making weight loss programs that relied on “selling a product”. Related to this was the altruistic motivation of being in a “research program” by which participants could ultimately “help others”. 

Approximately one-third of participants were motivated to enroll, upon learning that the program was *aimed specifically at men*. 


*It really appealed to me because it was a men’s program. That’s one of the things that’s clear about men’s health, is that there’s not a great deal of programs that people comfortably know about out there. And so, a lot of people, I believe, they just simply carry on because they don’t really know how to access it. And I liked the idea that it was also focused on the way that men think. It didn’t necessarily have to be run by men, but it was more comfortable. There’s nothing worse than trying to go to a place or join a program where it’s primarily women that are doing it.*
(ID 2, 58 years, BMI: 39.8 kg/m^2^, PHQ-9 score: 6) 


*I think it was aimed at men. I think that was the bit that got me over the line at the end. It was just purely aimed at men because a lot of guys are... What’s the word? I can’t even think of the word I want. We’re not a closet, but we’re stubborn, I think the word is, we are stubborn. And with this aimed at men I could get a bit more out of it. I think that was the main thing.*
 (ID 12, 58 years, BMI: 28.4 kg/m^2^, PHQ-9 score: 12)

Half of the respondents joined the program after hearing it *advertised* on the radio or on other platforms. The *gender-tailored messaging* of the project recruitment material encouraged motivation to engage with the program, with a few participants reporting the advertising was particularly salient for them and sparked a “lightbulb moment” of the need to make a change.


*I saw it on Facebook one night when I was feeling pretty vulnerable, I suppose. I was working away from home, on my own, and probably had too much to drink, and just feeling a bit flat. And every single policy, or whatever, you were advertising hit home to me. And so, I’m right well shit I’ve got to have a crack at that. It seemed almost too good to be true.*
 (ID 4, 37 years, BMI: 27.3 kg/m^2^, PHQ-9 score: 7)

While most wanted help with weight loss, just under half of the participants had been directly motivated to join due to their *mental health issues*, with the majority of these related to depression or anxiety. Many of those participants enrolled because of the program focus on the psychology of weight loss, in the hope of addressing mental health issues, and alluded to an existing developing sense of the interrelatedness of mental and physical elements. Hence, these participants saw *SHED-IT: Recharge* as a rare opportunity to join a program that not only explicitly acknowledged but also addressed both of these issues.


*Probably the thing that jumped out at me was addressing both the weight and the mental health, and probably both of those were relevant to me. I think I’ve been struggling with both of them, so that was where it was interesting just seeing the two being put together.*
(ID 19, 53 years, BMI: 28.7 kg/m^2^, PHQ-9 score: 5) 

### 3.4. Unique Opportunity to Implement Sustained Physical and Mental Health Changes Compared to Previous Experiences

Many men commented on the potential for *SHED-IT: Recharge* to offer a different experience for them compared to wither weight loss or mental health programs in their past. Approximately one-third of participants reported having used some of the better-known weight loss programs (e.g., Lite N Easy, Weightwatchers), consulted with dietitians or personal trainers, while a few others talked of self-driven attempts at weight loss.


*The disappointing thing is for me I’ve been going to a gym on and off for 20 years and the results were good, but I never sustained them because I hadn’t put two and two together.*
 (ID 1, 50 years, BMI: 41.6 kg/m^2^, PHQ-9 score: 11)

While many of those who had engaged in weight loss attempts in the past reported having achieved positive short-term results, these were not sustained with weight gain inevitable once the program had ceased or old habits were re-established.

Approximately one-quarter of participants voiced having consulted with a *mental health professional* for mental health issues, while a few others had attempted to address low mood on their own.


*As far as approaching anyone about mental state, no, that’s really ‘no go’ I’m afraid. I’m pretty [much], what you call, not closed but I keep things to myself.*
 (ID 15, 62 years, BMI: 32.2 kg/m^2^, PHQ-9 score: 13)

Interestingly, very few talked about how previous interventions had provided mental health benefits for them, and the longevity of these benefits.


*And then I got the shits with that because I just felt like I was repeating myself and I was boring myself with the same old shit, so I just said I’m out, thank you.*
(ID 4, 37 years, BMI: 27.3 kg/m^2^, PHQ-9 score: 7) 

### 3.5. Salience of the Program Elements

#### 3.5.1. Importance of the Handbook

While content presented in the handbook was a core element of the program that linked many of the sub-themes in this evaluation, specific mention should be made of participants’ discussion of the important impact of the handbook. A lot of discussion was had around the empowering value of knowing “how weight loss works”, and understanding the principles behind it, such as kJ input/output and resting metabolic rate.


*It was very much the handbook and the logbook. For me it all stemmed back to the resting metabolic rate. Once I accepted that, which was immediately, it’s oh that’s how to get slimmer. Once I accepted that, that then drove everything else.*
(ID 1, 50 years, BMI: 41.6 kg/m^2^, PHQ-9 score: 11) 


*I liken it to a mathematical equation. So, you’ve got energy input in here, energy consumed, and so long as there’s a bit of a deficit between the amount of energy that you consume and the amount you use, then that means that you’re going to continuously [achieve] weight loss.*
(ID 2, 58 years, BMI: 39.8 kg/m^2^, PHQ-9 score: 6) 

Indeed, it was the level of knowledge and understanding imparted that was felt to set the program apart from other programs with the same goal.


*And the difference is this time round I’ve got a better understanding of the nutrition side of things than I previously did.*
(ID 16, 58 years, BMI: 35.5 kg/m^2^, PHQ-9 score: 7) 

By many, the handbook was seen to facilitate informed choices that would form the basis for sustainable and permanent changes.


*From the program point of view, I think what the program did do was make you acutely aware of the nutritional elements in every type of food. And also reading the actual packaging and realizing that sucrose, dextrose and the whole trip of them is pretty much the same thing.*
(ID 3, 47 years, BMI: 41.2 kg/m^2^, PHQ-9 score: 10) 

#### 3.5.2. Importance of the Logbook

The logbook provided an important physical manifestation of progress for many with the process of charting and recording acting to help build good new habits.


*Because of the logbook, and recording things, having the triggers there that prompt you to do it and to build in those good habits. I have, just in the last week, set myself a new goal around losing weight while I’m doing a training program with some people from work.*
(ID 18, 43 years, BMI: 28.5 kg/m^2^, PHQ-9 score: 10) 

For a few participants, the logbook was seen as such an important part of their success that they had created their own logbook to keep the routine going after the program finished. 


*It gave me more of a purpose with regards to having a daily schedule. I still write stuff down. There’s my little notepad, that’s what I’m doing today, so I’m going to be purposefully making sure that I’ve got a target for the day.*
 (ID 16, 58 years, BMI: 35.5 kg/m^2^, PHQ-9 score: 7)

#### 3.5.3. Importance of the MyFitnessPal App

The MyFitnessPal app was overwhelmingly positively received by participants, many of whom reported still using it on a routine basis. While acknowledged to be one of an important set of tools, the app was an integral component of self-monitoring, which was perceived as crucial for success.


*That’s the MyFitnessPal one. I don’t mean that that would work all on its own, I mean as part of your overall program that was probably one of the most important things for me and it still is.*
 (ID 10, 63 years, BMI: 34.2 kg/m^2^, PHQ-9 score: 9)


*The best thing for me was using that MyFitnessPal app and physically tracking my calories was probably the most influential thing. It was the thing that I got most success out of.*
(ID 11, 59 years, BMI: 32.2 kg/m^2^, PHQ-9 score: 13)

Apart from the more obvious monitoring of kJ input/output that it facilitated, the app was described by many as aiding motivation and goal setting, as well as being at the core of food awareness, and that was clearly the most important function it had served.


*I think that’s the big thing with using MyFitnessPal, it’s the mindfulness about what you’re doing, not just mindlessly putting food in your mouth, but realizing what it is that you’re putting in and getting that overall balance is very powerful for being in control of your weight.*
(ID 18, 43 years, BMI: 28.5 kg/m^2^, PHQ-9 score: 10) 

#### 3.5.4. Importance of the Mental Fitness Modules

Most participants commented that the mental fitness modules had been helpful in providing strategies for dealing with “unhelpful” thoughts and for some, these modules had provided them with tools that were perceived to be on par with, or more helpful than, other sources of mental health support that they had accessed in the past.

The modules were mentioned as having been particularly helpful in averting a downward spiral of negative emotions or shifting the focus away from negative thoughts or dealing with daily stresses. The specific aspects of meditation and mindfulness were mentioned by several participants as helpful.


*With the mindfulness, when I am stressed and that sort of thing you can just take even ten minutes to practice that and it feels like it resets you a bit, like resets the stress level. It brings everything down and gets you a lot calmer, so that’s helped a lot.*
(ID 14, 28 years, BMI: 30.9 kg/m^2^, PHQ-9 score: 18) 

Others talked in more general and broad terms about the modules having helped them approach life and its challenges with a greater level of equanimity.


*They were quite good, and I did put in an effort, few different things. I don’t care if somebody cuts me off in the traffic anymore. I just think, well, tough. Things like that. [I] changed my focus on the way I react, rather than the way the others react.*
(ID 5, 62 years, BMI: 27.0 kg/m^2^, PHQ-9 score: 5) 

In contrast, approximately one-quarter of participants, had not engaged with the mental fitness modules, either from a perceived lack of need, or lack of time.


*I don’t know why but that didn’t really ring for me. I’m sure it did for many people but for me, and I really, I can’t remember. I think I was just busy at the time or there was a lot of work pressure on and maybe I didn’t pay enough attention to that.*
(ID 10, 63 years, BMI: 34.2 kg/m^2^, PHQ-9 score: 9)

#### 3.5.5. Importance of Resistance Training Videos

The resistance training videos were well received overall, with only two participants voicing that they had not engaged with them. They were described as having great variety of exercises, delivered in a simple and non-complicated manner, and importantly, not requiring special equipment.


*They were easy to do in the house, you can do them anywhere. I’d be away for work in the hotel room, and I’d be doing the airplanes and the twists and the planks and things like that. Mostly planks on the elbows because of my [sore] wrists. But I think the fact that you can do them anywhere was a big factor.*
 (ID 10, 63 years, BMI: 34.2 kg/m^2^, PHQ-9 score: 9)

It was particularly mentioned as beneficial that the range of exercises included in the videos were achievable, which also made some feel good about their progress. 


*You do get confidence from knowing that you can see yourself improving. They weren’t that complicated or hard to do either.*
(ID 8, 52 years, BMI: 28.9 kg/m^2^, PHQ-9 score: 5) 

Only a few participants suggested that they would have preferred to have different versions available so that they could choose according to their physical capabilities and fitness level.

#### 3.5.6. Importance of Online Delivery 

The vast majority of participants indicated that the online delivery made engagement achievable, enabling them to fit the program around their family and work commitments.


*Having an appointment would mean I would have to miss out, start late or finish early or whatever to get to those appointments. Online, I’ve got my free time and I can do whatever I need to do when I got time and not having to specify.*
(ID 5, 62 years, BMI: 27.0 kg/m^2^, PHQ-9 score: 5) 

It was also considered important to receive regular updates, reminders, and personal progress reports. A few preferred email notifications but most felt that their mobile phone was a better medium for this.


*You go through your day, and you can drift away from what your biggest goals and objectives are, those little things. You get an email, you get a notification, a reminder on your phone about, go and enter your food into MyFitnessPal, or what your goals are. That really works with me as a trigger and a reminder about what it is that I’m doing, to sort of bring you back to that, that you can drift from. Having it in an app, bring it up on your phone, and it’s easy. I thought it was great.*
(ID 18, 43 years, BMI: 28.5 kg/m^2^, PHQ-9 score: 10) 

### 3.6. Further Opportunities That Build Accountability could Help Maintain Focus

A range of suggestions for improvements were made beyond those that have been covered in the above sections. These included:

#### 3.6.1. Longer-Term Follow-Up

This included the availability of a longer-term formal program duration to the more commonly mentioned scheduled “check-in” subsequent to program cessation. The frequency of such scheduled check-ins ranged from monthly to annually. Some suggested a simple phone call, while others mentioned a weigh-in session, a motivational interviewing style session where participants could discuss difficulties and revised goal setting. The purpose of this continued contact with the program was for most seen as a way to maintain a level of accountability and motivation. 

#### 3.6.2. More Frequent and Personalized Contact with Program Staff 

This and the following recommendation, both centered largely on the facilitation of a “*SHED-IT: Recharge* community spirit”. More regular check-in by program staff during the program was one aspect of this. Another suggestion was more regular program updates that could include participants’ success stories.


*You could even just throw it out, if there’s a weekly update, and it’s [a] share your success stories, people put in, I did this, I ran 3kms this week, and I couldn’t do that before. Just those things that might resonate with someone and reinforce that to them.*
(ID 18, 43 years, BMI: 28.5 kg/m^2^, PHQ-9 score: 10) 

#### 3.6.3. Facilitation of “Peer Support”

One of the most predominant suggestions for improvement centered on peer support: the creation of a “community” for the provision of support, and the sharing of ideas and successes. Some of the suggested venues for this were online Zoom meets, Facebook groups, or buddy systems.


*It also gives them another contact group as well…if anyone’s struggling, a lot of times you’ll find that some people will open up in a group. Some people won’t, but if it’s there and it’s positive, and you all want to get together, it would be quite good to do.*
(ID 16, 58 years, BMI: 35.5 kg/m^2^, PHQ-9 score: 7) 


*Well, I realize obviously it’s about time, and money, and how much funding you can get, or whatever. But even a Zoom meeting, you could have 20 blokes or 30 blokes on there at a time and then you’re accountable I reckon. You’ve made a commitment. It’s the same time every week or fortnight, or whatever it is. You make a plan and say right at six o’clock, or eight o’ clock or whatever, we’re going to get together. And we’re going to first of all do the exercise and then we’re going to have a yak or something. And then I’m right-o well shit I haven’t done my homework I’d better do it. That’s how I operate personally.*
(ID 4, 37 years, BMI: 27.3 kg/m^2^, PHQ-9 score: 7) 

#### 3.6.4. More Interactive Features

Some suggestions were made relating directly to the delivery of the program. Quite a number of participants would have valued a more interactive delivery mode, taking advantage of smartphone capabilities, as these were considered “harder to ignore”.


*It was not hard to do but it’s easy to leave emails and, I’ll go look at them on Wednesday arvo…Email’s a bit easier to brush, I suppose. Or just go, I’ll look at that later, whereas if something just pops up and it’s not like I’ve got to go in and read anything and find any information. It just comes up and says, “Have you done it?” Or “This is what you’ve done”. That kind of thing. Or “Your workout’s due today”.*
(ID 9, 33 years, BMI: 34.5 kg/m^2^, PHQ-9 score: 8) 

A few also expressed having preferred more frequent (daily) personal and progress information sent via phone/text. One participant felt that the use of additional questions and prompts would have increased accountability and engagement. 


*If it’s a question about “Did you record your food today?” or “Did you get enough sleep last night?”, or “Have you done your activities” or “This is your day to step on the scales reminder” might just be the push that I needed to be held accountable.*
(ID 9, 33 years, BMI: 34.5 kg/m^2^, PHQ-9 score: 8) 

While many had valued the flexibility inherent in a predominantly online program, some would have preferred the availability of more face-to-face and personalized contact.


*And as much as I want to, or need to, or whatever, it doesn’t work for me, I’m more of a tangible face-to-face sort of person. And I think I connect better if there’s perhaps more face-to-face interaction getting to know people rather than just material.*
(ID 4, 37 years, BMI: 27.3 kg/m^2^, PHQ-9 score: 7) 

## 4. Discussion

This study explored the experiences of men who participated in the *SHED-IT: Recharge* program. The findings contribute novel insights to what is currently only a small body of research investigating the impact of behavior change programs comprising specific mental health support to engage and assist men living with elevated depressive symptoms and weight status. Four key areas of focus emerged to provide insights for the unique and important role such gender-tailored programs offer to improve the physical and mental health of men with low mood. These analyses provide real-life perceptions outlining why men chose to enroll in the program, their high levels of satisfaction with the program, and suggestions for future program improvements. 

In summary, the primary goal for men enrolling in the program was to receive weight loss support and guidance to improve health behaviors. This appears consistent with process evaluation data from previous gender-tailored behavior change interventions [36,47,48]. While some men were encouraged to participate by their partner, men also indicated they were attracted to the specific gender-tailoring of the recruitment materials and the credibility of the program as a university-based intervention. These reasons support enrolment insights from other weight loss trials targeting men [36,49], which highlights men are willing to participate in programs that are adapted to align with their preferences and values (e.g., gender-tailored, credible sources) [14,17]. A noteworthy finding was that prior to enrolling in the *SHED-IT: Recharge* trial, some men revealed seeking mental health support was a “no go” because of difficulty expressing emotion. This aligns with other evidence suggesting men can have difficulty seeking help to treat mental health concerns [9,50]. In this context, it was of interest that many were attracted to the integrated mental and physical elements the program offered to decrease weight and improve mood simultaneously. The men’s participation and acceptability of the program was also positive, including novel components that explicitly targeted mental health (i.e., online mental fitness modules). This provides vital evidence for the utility of eHealth programs targeting men with low mood to include practical mental health strategies that alleviate their depressive symptoms, particularly for men who favor confidential and autonomous support [51,52]. These positive experiences may increase the potential for men to access the “therapeutic front door” to more specialized mental health services when in heightened emotional states [53]. 

Men’s perceptions revealed they were very satisfied with the *SHED-IT: Recharge* program and its components. Specifically, men appreciated the evidence-based information presented in the handbook and the applied logbook tasks focusing on education and sustainable strategies to create behavior change. This aligns with other gender-tailored lifestyle programs where men valued practical and explicit messages (e.g., visualizing recommended portion sizes, useful physical activity advice) to facilitate weight loss [47,54]. Although physical activity is commonly targeted in behavior change programs, it was promising that the novel online resistance training videos were well-received. Notably, with a previous self-directed study for men that did not include videos reporting difficulty when targeting resistance training (e.g., lack of familiarity, knowledge) [49], this suggests videos that demonstrate exercises (including variations) and guide men through the session may be important for increasing engagement. Additionally, the low-dose nature of the program provided men with relative simplicity, yet they identified that the intervention components kept them accountable and motivated. This finding is comparable to previous *SHED-IT Weight Loss* process evaluations, which have also emphasized the program may offer a cost-effective and convenient approach to help men achieve weight loss at a population level [32,36]. Importantly, eHealth programs can enhance scalability by directly addressing barriers that decrease the practical translation of face-to-face interventions (e.g., multiple consultations, ongoing costs) [22].

Although the men valued the self-directed format of *SHED-IT: Recharge*, the need for increased social support was raised by a subgroup of men. A future study may compare an online intervention with and without the social connectedness element to test whether this results in a greater impact for treating depression (balanced against the increased program costs). Of interest, a recent qualitative study exploring mental health promotion in the workplace reported men perceived social cohesion as particularly meaningful for “having each other’s backs” [55]. While evidence suggests men appreciate health programs that provide independence and confidentiality [19,24,36], consideration of treatment preference in relation to delivery modality as a future model may be warranted for this subgroup. For example, with eHealth programs constantly evolving, future studies could consider using interactive approaches that integrate physical and mental health support for men who would prefer greater intervention contact. A possible strategy may include a regular online session that begins with targeted exercises or behavior change advice followed by an opportunity for men to discuss their achievements or concerns relating to physical and/or mental health. Another important suggestion for program refinement was to have the content presented within an app to improve functionality and availability. Indeed, this concept may increase the likelihood of meaningful program dissemination, which could be easily accessed by men in regional and remote areas who often have limited access to professional health services [56].

To our knowledge, this is the first qualitative study to explore men’s perceptions of a gender-tailored lifestyle program targeting both the physical and mental health of men. In particular, these data provide a rich understanding evaluating the experiences of men living with both low mood and overweight or obesity who are widely acknowledged as an underrepresented subgroup in the psychological and weight loss literature [10,52]. This study had several strengths including the stratification of men by baseline depressive symptoms and BMI using the RCT dataset to provide a combined sample of men impacted by varying categories of depressive symptoms (i.e., mild/moderate-to-severe) and weight (i.e., overweight/obesity). Further, an experienced qualitative researcher (V.H.) gave careful consideration to the systematic organization and analysis of the data. The study also has some limitations to acknowledge. First, although the program was self-directed, the interviewer was a member of the research team and made brief contact with the men during assessments. While we did not have the capacity for an external party to conduct the interviews in the current study, this would be worthwhile in future studies to reduce the potential of social desirability bias in participants responses. In order for this potential bias to be mitigated in the current study, our thematic analysis was conducted by an external qualitative researcher (author V.H.) who was not involved in any other aspects of the study. Second, as men were invited to participate in an interview at the final assessment, the experiences of the men not retained in the study may not be represented in the current analysis. In future studies, it would be valuable to seek men’s consent for a potential interview at the beginning of the trial, or to provide participants with an optional exit interview when they disengage from the trial. Third, although men were specifically targeted in the intervention (i.e., gender-tailored), many diverse subgroups of men exist (e.g., age, masculinities, social class, cultural background, ethnicities, sexual orientation). In future studies it will be important to investigate whether these findings are transferable to all men or only those from particular subgroups, however this was beyond the scope of the current study. Finally, given our intervention simultaneously targeted men’s physical and mental health, it is somewhat difficult to determine which components were most appealing or which led to the largest health improvements. In order for some of these more nuanced questions to be teased out, it would be valuable to compare men’s engagement and outcomes when randomized to comparable physical vs. mental health interventions in future research.

## 5. Conclusions

In conclusion, the current process evaluation revealed that although men are traditionally hard to engage in both weight loss and depression research [8,9], they were willing and interested to participate in a behavior change program with integrated mental health support designed specifically for them. The self-directed nature was appealing and engaging because it provided convenience and flexibility, but greater program support (e.g., from both researchers and peers) may be needed for some men to maintain focus. The intervention components were not perceived as resource-intensive but were sufficient to motivate men with low mood to implement meaningful behavior changes that enhanced their physical and mental health. Some important suggestions to improve the program were made, including the development of an app to store content in a central location. Importantly, this would provide increased access and highlights strong potential for realistic implementation on a large scale; however, this requires further testing in trials that specially target translation. Overall, study findings provide in-depth insights into men’s experiences of the program, which indicate gender-tailored, self-directed lifestyle interventions comprising mental health support are acceptable and satisfying for men living with depression and overweight or obesity.

## Figures and Tables

**Table 1 ijerph-18-12878-t001:** Semi-structured interview guide.

SHED-IT: Recharge was pitched as a program that would try to help men lose weight and improve their mood. When you registered for the study, what were your main concerns that you wanted help with? Before this study, had you ever sought help to lose weight or improve your mood? Was there anything about this study in particular that influenced your decision to enrol? Have you noticed any changes in your weight or physical health you feel are related to the SHED-IT: Recharge program? Have you noticed any changes to your mood that you feel are related to the SHED-IT: Recharge program? Have you made any changes to your physical activity levels as a result of participating in the program? Have you made any changes to your eating habits as a result of participating in the program? Outside of physical activity and diet, did you make any other changes during the program that you feel had an impact on your weight loss or mood? (e.g., alcohol, sleep, other drugs). In addition to targeting physical fitness, the program included strategies targeting mental fitness (e.g., managing unhelpful thinking patterns and stress, mindfulness). What were your experience like with these aspects of the program? In your opinion, what is the one key thing that you got from the SHED-IT: Recharge program that you will keep doing or thinking? What was the best outcome of the program for you? Do you have any suggestions that could improve the program? Anything else you would like to add that we haven’t covered?

**Table 2 ijerph-18-12878-t002:** Baseline characteristics of men from the *SHED-IT: Recharge* intervention group.

Characteristic	(*n*, %)		
	Interviewed (*n* = 19)	Not Interviewed (*n* = 43)	Total (*n* = 62)
Age (mean, SD)	49.6 (11.6)	46.1 (11.7)	47.2 (11.7)
PHQ-9 score	9.0 (3.7)	9.2 (4.4)	9.1 (4.2)
PHQ-9 category			
*Mild (5–9)*	10 (53)	26 (60)	36 (58)
*Moderate-to-severe (≥10)*	9 (47)	17 (40)	26 (42)
Existing mental health support	8 (42)	13 (30)	21 (34)
Weight (kg)	103.4 (17.0)	104.3 (14.9)	104.1 (15.5)
BMI (kg/m^2^)	32.5 (4.6)	32.5 (3.5)	32.5 (3.8)
BMI category			
*Overweight (BMI 25–29.9 kg/m^2^)*	8 (42)	10 (23)	18 (29)
*Obesity (BMI 30–<42.0 kg/m^2^)*	11 (58)	33 (77)	44 (71)
Born in Australia ^a^	17 (90)	38 (88)	55 (89)
Married/domestic relationship	14 (74)	35 (82)	49 (79)
Post-school qualifications	17 (90)	38 (88)	55 (89)
Employment status			
*Employed full-time*	13 (68)	23 (53)	36 (58)
*Employed part-time*	2 (11)	1 (2)	3 (5)
*Self-employed*	1 (5)	5 (12)	6 (10)
*Retired/not seeking work*	2 (11)	6 (14)	8 (13)
*Unemployed or other*	1 (5)	8 (19)	9 (15)
Socio-economic status quintile			
*1 (Most disadvantaged)*	0 (0)	4 (9)	4 (6)
*2*	6 (32)	8 (19)	14 (23)
*3*	7 (37)	23 (53)	30 (48)
*4*	4 (21)	6 (14)	10 (16)
*5 (Most advantaged)*	2 (11)	2 (5)	4 (6)

^a^ India (*n* = 2); Germany, New Zealand, South Africa, United Kingdom, Yugoslavia (*n* = 1).

**Table 3 ijerph-18-12878-t003:** Themes and sub-themes emerging from the interview with men participating in the SHED-IT: Recharge program.

Theme	Sub-Theme
Specific circumstances determined men’s motivation to enrol	Dissatisfaction with weight Prompted by partner University-based affiliation Gender-tailored advertising and content Opportunity to include mental health
Unique opportunity to implement sustained physical and mental health changes compared to previous experiences	Previous weight loss interventions Previous mental health interventions
Salience of the program elements	Handbook Logbook MyFitnessPal Mental fitness Resistance training Online delivery
Further opportunities that build accountability could help maintain focus	Longer-term follow-up More frequent personalised contact with staff Peer support More interactive features

## Data Availability

The data presented in this study are available on request from the corresponding author. The data are not publicly available due to confidentiality reasons.

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
