# Peer review of "Men’s Perceptions of a Gender-Tailored eHealth Program Targeting Physical and Mental Health: Qualitative Findings from the SHED-IT Recharge Trial"

_ijerph, 2021, doi:10.3390/ijerph182412878_

Round 1
Reviewer 1 Report
This study is well-designed and written, although it presents only qualitative data. The findings of this research are an example of strategies aiming at dealing with the increasing burden of both mental disorders and obesity.
I did not find any particular flaws in this manuscript. I checked the comments of the other reviewer and they seem valid. In my opinion, since the article has a descriptive character, it should be published as a short communication rather than research article.Author Response
Reviewer 1
Comment:
This study is well-designed and written, although it presents only qualitative data. The findings of this research are an example of strategies aiming at dealing with the increasing burden of both mental disorders and obesity.
I did not find any particular flaws in this manuscript. I checked the comments of the other reviewer and they seem valid. In my opinion, since the article has a descriptive character, it should be published as a short communication rather than research article.
Response:
Thank you for your positive feedback We appreciate you taking the time to review our paper. To confirm, although our paper includes participant quotes and reflections, these were representative of underlying themes that emerged during a rigorous qualitative research process. Further, as this is the first qualitative study to explore men’s perceptions of a gender-tailored lifestyle program targeting both the physical and mental health, we strongly believe that the novel findings presented in this paper make an important contribution to a consistently underrepresented subgroup in weight loss and depression research.
Reviewer 2 Report
Thank you for giving me the opportunity to read your manuscript “Men’s Perceptions of a Gender-Tailored eHealth Program Targeting Physical and Mental Health: Qualitative Findings from the Shed-It Recharge Trial.”
I think this is an important study on the evaluation of a gender-tailored intervention for men who suffer from obesity and depressive symptoms. The understanding of what men motivated to participate, how they perceived the intervention, and what they would change can help with implementations of such and similar programs. Especially positive is the gender sensitive approach and suggested strategies how to enable men to use mental health services but still be able to uphold certain masculinity ideologies.
I have some concern and suggestions how the manuscript could be improved. My main concerns are summarized and followed by detailed suggestions:
I) More details needed on the “gender-tailored” aspects of the intervention.
II) Base your interview-development on literature.
My detailed concerns and suggestions:
Major Points
Abstract
1) Good and informative abstract. Maybe mention in a short statement in how far the intervention is “gender-tailored” and what exactly it addresses (and also what is “e-“ electronic about it).
Introduction
2) A very engaging Introduction. I would recommend being more specific what did “gender-tailored” interventions specifically include in addition to “conventional” therapies/approaches (line 56)
3) I suggest defining “eHealth” (line 64). Also explain why men would prefer eHealth options (what traditional masculinity ideologies could be easier uphold?).
4) Be more specific on and especially offer details on the “gender-tailored” aspect of your intervention (line 68). How exactly did the intervention look like, what did it include? How long/how many sessions was the intervention? What did you/men do?
The SHED-IT program is described in line 138 and following (Methods point 2.3). I suggest mentioning and describing the program earlier, e.g., first chapter of the Methods section or at the end of Introduction.
Be also sure to mention clearly all later brought up components. E.g., the “resistance videos” are highlighted by the participants as being a main component, but we (readers) do not know much about them.
5) “Compared to the control group” (line 74) – control group was men not receiving treatment/waiting list/other treatment?
6) Maybe annotate as “ΔM = -2.4 score points at the Patient Health Questionnaire (PHQ-9), 95% CI = -4.0 – -0.7, p<0.01, d = 0.5” and “ΔM = -3.6 kg, 95% CI = -5.1 – -2.1, p < 0.001, d = 0.8” (line 76-77)
7) Report the source again after stating the detailed results of the already published studies. (line 78-79).
Methods
8) I do not understand “six months after completing the SHED-IT: Recharge trial (primary time-point was three months).” (line 99). Interviews took place six months after the intervention. What is the “primary time-point”?
9) How many men participated in the intervention? (line 122) How many of those who participated are the selected 46 men; who agreed to participate in the interview?
10) What method/procedure/software was used to select randomly (line 128)?
11) How was the sample size of 20 chosen/determined?
12) Describe in the Study Design what did “online” entail, video-chat/conference – report Zoom earlier than line 224? Were video-conferences recorded (audio and video)?
13) Base your interview-development on literature. E.g., refer to guidelines or previous evaluation studies that report on how to assess the specific indicators for your evaluation of the program.
14) “As seen in Table 1, some questions also explored whether the program had an effect on men’s physical and/or mental health, however these findings will be addressed in a subsequent paper.” Line 204, Were responses to those questions also used to inform about “program appeal, program characteristics, experiences, satisfaction and the impact of the SHED-IT: Recharge program” (line 202)? If yes, this statement (in line 204) is confusing. If not, maybe mark which questions specifically were considered in the current analysis.
15) What quantitative data was assessed?; line 217 states that quantitative data analysis was performed
16) The research question was the basis for coding the themes (“during the subsequent coding cycle, all data relevant to answering the overall research questions were coded” line 231). However, I have difficulty finding the spelled out research question. In the Introduction the study sought out to find out (line 89-91): “perceptions of men […] in relation to recruitment, satisfaction with program content and delivery components, and suggestions for […] program could be improved”. In the Methods the goals seem slightly different: “program appeal, program characteristics, experiences, satisfaction and the impact of the SHED-IT: Recharge program” (line 202). Be more detailed and explicit about your research questions in the Introduction and be consistent throughout the manuscript.
17) Report in the Methods section what sociodemographic characteristics and with what questions (+ response options) have been assessed. Also include information on the validated scales used (e.g., PHQ-9 [include psychometric descriptions]) and weight assessment.
18) What is a de-facto relationship? What relationship status did other men have? Where were other men born if not in Australia? Also express the other response options and numbers (not only the most prevalent).
Results
20) Can a number be reported, on how many statements or how many men made a statement related to a certain theme? This information could be added in Table 3 – so that the relative relevance of each statement/theme becomes evident. E.g., how many of the 19 men were motivated to enroll in the program because they were dissatisfied with their weight?
21) Maybe “implement sustained physical and mental health changes” instead of “create sustained physical and mental health changes” (theme heading)
22) prompted by their partner (line 281) – what gender did partners have?
23) also include the source participants for short citations such as “selling a product” (line 286)
24) The concept “advertisement” (line 303) does not seem like a motive. Maybe this category can be re-labeled to better describe the underlying motive. Also the category “mood or mental health issues” can be re-labeled to more transport the information that it seemed important to participants that “mood and weight related issues” (314) were addressed (mood seems to be part of mental health; thus the current label does not include weight).
25) Try to avoid paragraphs that consist of only one sentence (e.g., line 326)
Discussion
26) Offer a deeper discussion of your limitations and be more reflective in this part. Additionally, you may discuss how to address the limitations in future studies or how limitations might have impacted results.
27) “acceptable and satisfying for men living with depression” (line 625) this statement and some others in the manuscript highlighted only the depression of men. However, I think that this is inaccurate as men were also unsatisfied or having problems with their weight. It might also be interesting whether the seeable body problem or the (easy to hide) mental health problem was the “justification” for men to participate. Maybe combining mental health with physical health might help address also other metal health problems. Irrespective of whether you pursue this though, I recommend staying precise when stating about whom the study was about, thus, do not depict men as only having depression, but depression and problems with weight throughout the manuscript.
Author Response
Please see the attachment. Thank you for your insights.

Reviewer 3 Report
Congratulations to the authors.
The paper deals with a timely and relevant topic, especially in considerations of these COVID times.
The paper is well and accurately presented in all of these section.
Only one question: participants were interviewed during COVID emergency? If yes, do the authors think it could have affected participants retrospective perceptions on the study (and then have impacted on results from this study)? And, if yes, how? They could add this consideration in the discussion section.
The study aims at collecting insights for future program development and research regarding programs for men with obesity and mood disorders. Particularly, through qualitative methods, it aims at identifying aspects of the program that participants could find engaging and effective, in order to improve efficacy of prevention programs.
The topic of programs for men with obesity and mood disorders has already been investigated. However, this is the first time that qualitative method apply to the field. These analyses provide real-life perceptions outlining why men chose to enroll in the program, their high levels of satisfaction with the program, and suggestions for future program improvements.
It is well described in the first and final part of the discussion. This is the first qualitative study to explore men’s perceptions of a gender-tailored lifestyle program targeting both the physical and mental health of men. The findings contribute novel insights to what is currently only a small body of research investigating the impact of behavior change programs comprising specific mental health support to engage and assist men living with elevated depressive symptoms and weight status. THe study shows the unique and important role such gender-tailored programs offer to improve the physical and mental health of men with low mood.
The methodology is adequate, detailed, and well described.
The study sample is limited, however it is sufficient for a qualitative study.
The conclusions are consistent with the evidence and arguments presented. They address the main question posed,
The paper is adequately referenced.
Tables and figures are clear and informative.
Author Response
Reviewer 3
Comment:
Congratulations to the authors.
The paper deals with a timely and relevant topic, especially in considerations of these COVID times.
The paper is well and accurately presented in all these sections.
Only one question: participants were interviewed during COVID emergency? If yes, do the authors think it could have affected participants retrospective perceptions on the study (and then have impacted on results from this study)? And, if yes, how? They could add this consideration in the discussion section.
Response:
Thank you for your positive feedback, we appreciate your valuable insights and taking the time to review our paper. We can confirm that the interviews took place during the very early stages of the COVID pandemic. However, we specifically asked men to discuss their experience within the intervention time points. We have addressed this in the manuscript on line 176 as follows:
“Although men were interviewed during the very early stages of the COVID-19 pandemic, participants were specifically asked to reflect on their experiences during the program and the subsequent follow up period (which occurred prior to the pandemic).